# Identification of Larch Caterpillar Infestation Severity Based on Unmanned Aerial Vehicle Multispectral and LiDAR Features

**Sa He-Ya** [1,†]**, Xiaojun Huang** [1,2,3,*,†]**, Debao Zhou** [4]**, Junsheng Zhang** [4]**, Gang Bao** [1,2]**, Siqin Tong** [1,2]**, Yuhai Bao** [1,2]**, Dashzebeg Ganbat** [5]**, Nanzad Tsagaantsooj** [5]**, Dorjsuren Altanchimeg** [6]**, Davaadorj Enkhnasan** [6]**, Mungunkhuyag Ariunaa** [5] **and Jiaze Guo** [1]

1   College of Geographical Science, Inner Mongolia Normal University, Hohhot 010022, China; 20224016050@mails.imnu.edu.cn (S.H.-Y.); baogang@imnu.edu.cn (G.B.); tongsq223@imnu.edu.cn (S.T.); baoyuhai@imnu.edu.cn (Y.B.); 20224016028@mails.imnu.edu.cn (J.G.)
2   Inner Mongolia Key Laboratory of Remote Sensing & Geography Information System, Hohhot 010022, China
3   Inner Mongolia Key Laboratory of Disaster and Ecological Security on the Mongolia Plateau, Hohhot 010022, China
4   Forest Bidogical Disaster Prevention and Control (Seed) Station, The Great Khingan Montains of Inner Mongoli, Yakeshi 022150, China; zhoudb321@163.com (D.Z.); cebao110@126.com (J.Z.)
5   Institute of Geography and Geoecology, Mongolian Academy of Sciences, Ulaanbaatar 15170, Mongolia; ganbat_d@mas.ac.com (D.G.); tsagaantsoojn@mas.ac.mn (N.T.); mungunkhuyaga@mas.ac.mn (M.A.)
6   Institute of Biology, Mongolian Academy of Sciences, Ulaanbaatar 13330, Mongolia; altanchimeg_d@mas.ac.mn (D.A.); enkhnasand@mas.ac.mn (D.E.)
*   Correspondence: hxj3S@imnu.edu.cn
†   These authors contributed equally to this work.

**Abstract:** Utilizing UAV remote sensing technology to acquire information on forest pests is a crucial technical method for determining the health of forest trees. Achieving efficient and precise pest identification has been a major research focus in this field. In this study, *Dendrolimus superans (Butler)* was used as the research object to acquire UAV multispectral, LiDAR, and ground-measured data for extracting sensitive features using ANOVA and constructing a severity-recognizing model with the help of random forest (RF) and support vector machine (SVM) models. Sixteen sensitive feature sets (including multispectral vegetation indices and LiDAR features) were selected for training the recognizing model, of which the normalized differential greenness index (NDGI) and 25% height percentile were the most sensitive and could be used as important features for recognizing larch caterpillar infestations. The model results show that the highest accuracy is $\text{SVM}_{\text{VI+LIDAR}}$ (OA = 95.8%), followed by $\text{SVM}_{\text{VI}}$, and the worst accuracy is $\text{RF}_{\text{LIDAR}}$. For identifying healthy, mild, and severely infested canopies, the $\text{SVM}_{\text{VI+LIDAR}}$ model achieved 90%–100% for both PA and UA. The optimal model chosen to map the spatial distribution of severity at the single-plant scale in the experimental area demonstrated that the severity intensified with decreasing elevation, especially from 748–758 m. This study demonstrates a high-precision identification method of larch caterpillar infestation severity and provides an efficient and accurate data reference for intelligent forest management.

**Keywords:** UAV multispectral; airborne LiDAR; *Dendrolimus superans (Butler)*; machine learning; pest severity

## 1. Introduction

The Larch Caterpillar (*Dendrolimus superans (Butler)*) belongs to the Lepidoptera, the family of leaf moths, and is widely distributed in Inner Mongolia, the three northeastern provinces of China, and the northern region of Xinjiang [1]. The insect is a two-year generation of pests that nibble on needles and leaves; the outbreak will eat all the needles and leaves so that the branches and trunks appear to look like after a fire, and when it is serious, it will lead to a part of the forest area to die [2], mainly harming the larch, the

pine, red pine and so on [3]. The Great Khingan forest area is a key carbon storage base in China [4] and is known as the "Green Great Wall of the Northern Border" [5]. The main tree species are larch, pine, birch, etc., of which larch forests are one of the top communities in the region, with larch accounting for about 86.1% of the forest area [6]. It was reported that in 2022, the insects occurred in approximately 8.9 hectares of the forested area of Great Khingan, Inner Mongolia, with an overall increasing trend. Of this total, 3.0967 hectares had mild infestations, 3.3007 hectares were moderately infested, and 2.5027 hectares were severely infested [7]. Therefore, there is an urgent need to utilize advanced technical means to take effective preventive and control measures for the larch caterpillar risk areas and provide technical support to green development.

Monitoring forest pests and diseases is usually based on traditional ground surveys, which can accurately quantify the physiological and biochemical components as well as the severity of infestation on a single tree, but this method has the disadvantages of being time-consuming, costly, and difficult to conduct at the scale of forests [8]. In recent years, with the booming development of UAV remote sensing technology, this approach has provided a time-sensitive [8], low-cost, and efficient method for forest pest monitoring and prediction, and remote sensing data with different spatial and temporal resolutions can be acquired even in large or complex forest areas [9,10]. To date, UAV hyperspectral and multispectral data are still the main data sources in forest pest monitoring and prevention [11], and the methods developed for constructing monitoring models with the aid of machine learning and deep learning are gradually maturing. Hyperspectral data are divided into imaging and non-imaging spectra, and their common characteristics are the large number of bands, the large amount of data in a certain spectral range to obtain continuous reflectance information, and the flexibility to use spectral information to integrate more spectral features, which can be applied to various types of pest research, such as via differential spectral reflectance (DSR), modified spectral indices (MSIs), and the continuous spectral index obtained from the continuous wavelet transform of DSR. Continuous wavelet coefficient of differential spectral reflectance (DSR-CWC) obtained by wavelet transform to monitor the leaf loss rate and estimation of population density under Erannis jacobsoni Djak (EDJ) damage showed that DSR-CWC had a significant sensitivity to the leaf loss rate, while DSR and MSI could adequately capture the reflectance features that were sensitive to population density [12,13]. In the early monitoring of pine wilt caused by pine nematodes, an accuracy of 88.11% was achieved by raw hyperspectral data combined with the 3D-ResCNN model [14]. Hyperspectral data are characterized by high cost, large data volume, and high hardware requirements in processing due to its hundreds of spectral bands. In contrast, multispectral data typically consist of 3 to 10 bands, making it advantageous due to its low cost, small data volume, simple processing, and quick access to useful spectral features. As a result, it is widely used in forestry pest monitoring. For example, different researchers used UAV multispectral data to monitor the extent of forest damage caused by different insect infestations (*Ips typographus*, L.) and different diseases (Areca yellow leaf disease). By calculating vegetation indices and texture features and constructing a backpropagation neural network (BPNN), Detection Tree, naive Bayes, support vector machine (SVM), and K-nearest neighbor (KNN) classification models. The final monitoring accuracy ranged from 86.30% to 90% [15,16]. Additionally, to classify pine wilt severity, a combination of unmanned aerial multispectral (MSI) and thermal infrared band (TIR) data using machine learning models (RF, SVM, 2D/3D-CNN) yielded that MSI was more effective for monitoring the healthy yellow and grey phases and that MSI and TIR performed the best for all infection phases (OA: 94.26%) [17]. Remote sensing images mainly refer to optical images from aerial images (multispectral and hyperspectral imaging data, etc.) and satellite images (Sentinel, Landsat, etc.), and, thus, can only provide a large amount of reflectance information in the horizontal direction of the forest trees, whereas light detection and ranging (LiDAR) can penetrate the canopy to provide the vertical structural characteristics of forests, and the combination of the two can adequately extract the three-dimensional structural information and spectral characteristics of the forests

and increase the accuracy of the monitoring of forest pests and diseases. For example, different researchers applied airborne LiDAR, airborne hyperspectral LiDAR (AHSL), and hyperspectral (HI) data to monitor forest pests (pine shoot beetle (PSB)) and diseases, where 3D radiative transfer models were used to extract canopy shading, calculate vegetation indices and LiDAR features to monitor stress with the help of the random forest algorithm. The potential of AHSL in predicting pest and disease stress was found to be in the range of 65.95%–89.45% overall accuracy, whereas, for PSB stress, LiDAR ($R^2 = 0.69$) accuracy was higher than HI ($R^2 = 0.67$) [18,19]. Dash J.P. et al. used manned aircraft and unmanned aerial vehicles to collect multispectral and LIDAR data to monitor invasive alien trees through random forest and generalized classification and regression models and found that all four classifiers and data combinations had Kappa values greater than 0.96 [20]. Most of the research in fusing UAV LiDAR and multispectral data are used for biomass estimation, tree height extraction, index inversion, tree species classification, etc., with fewer studies related to pest identification [21–24].

In summary, the red, red-edge, and near-infrared bands of UAV multispectral data are more sensitive to vegetation health and can quickly extract sensitive features to reveal the severity of larch caterpillar infestation [25–27]. UAV LiDAR is good for measuring forest structural characteristics and has a strong correspondence with the vertical profile of trees [28], so its combination with multispectral data will be advantageous to obtaining the canopy reflectance in the horizontal direction of a single plant, and the structural information in the vertical direction and the identification accuracy can be improved. Based on this, this study will use UAV multispectral LiDAR data and ground survey data as data sources, combined with a machine learning model to recognize healthy, mild, and severe larch caterpillar infestations to solve the following problems:

1.  Reveal the most sensitive feature sets (multispectral vegetation index and LiDAR features) of larch caterpillar infestations;
2.  Construct a high-precision model for effectively recognizing the severity of larch caterpillar infestation;
3.  Map the distribution of the severity of larch caterpillar pests on a single-plant scale in the experimental area and characterize their spatial distribution according to topographic features.

## 2. Materials and Methods

### 2.1. Study Area

The experimental area is located in a typical larch area approximately 4.62 km northeast of Wuerqihan Town, Yakeshi City, Hulunbeier, Inner Mongolia (Figure 1). The geographic coordinates of the center are 121°26′42″ E, 49°36′39.6″ N. It is rectangular in shape, with a length of approximately 922.691 m, a width of approximately 361.452 m, and an area of approximately 331,641 m². The region belongs to the cold temperate semihumid forest area and is a continental monsoon climate forest, with its main tree species being larch.

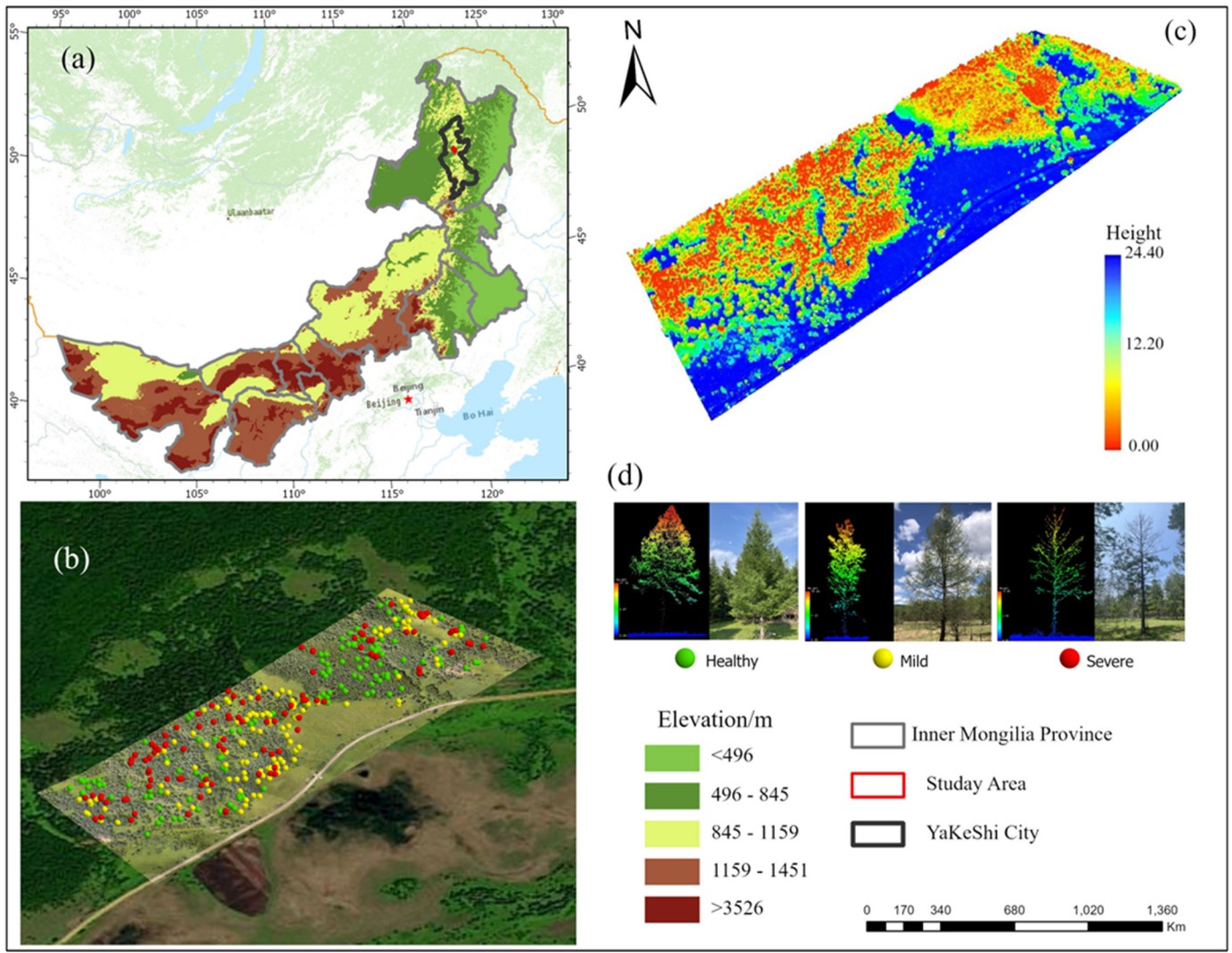

**Figure 1.** General map of the study area: (**a**) Spatial distribution of UAV multi-spectral data and sample tree; (**b**) UAV LiDAR point cloud data; (**c**) Field photographs and LiDAR profiles of healthy, mild, and severe sample trees; (**d**) Inner Mongolia digital elevation Model DEM.

*2.2. Data Acquisition*

2.2.1. Ground Survey Data

Ground survey data collection was conducted on 19 July 2021, using an A10 handheld GPS to obtain the geographic location of the larch; the device's positioning technology was provided by BDS + GPS + GLONASS; the first positioning time was 30 s; AGNSS positioning time was 5 s; the data updating rate was 1Hz, and a single-point positioning was 5 m. A tape measure was used to measure the east–west and north–south crowns and diameter at breast height (DBH). The leaf loss rate (LLR) was used to classify the severity of larch infestation. It refers to the ratio of the amount of canopy foliage loss per unit area to the amount of all foliage [13], which was calculated by selecting a typical standard branch in the east, south, west, and north directions of the sample tree at the upper, upper middle, middle, lower middle, and lower canopy levels, respectively, recording the number of injured needles and healthy needles and, finally, taking the average of the foliage loss rate of all the branches as the LLR of the current sample tree. A total of 356 samples were selected, of which 127 were healthy, 125 were mild, and 104 were severe. As seen in Figure 1c, the healthy forest trees have a green canopy with dense needles; the leaf loss rate is in the range of 0%–5%, and the point cloud density is higher from the LiDAR profiles; the canopy color of the mildly damaged forest trees ranges from green to yellow; the needles are sparse;

the leaf loss rate is in the range of 6%–15%, and the point cloud density is higher in the upper canopy than the lower; the canopy color of the severely damaged forest is grey; the branches are mostly exposed, and the leaf loss rate is in the range of 16%–100%, and from the LiDAR profile, the forest as a whole has only branches and only a few needles.

### 2.2.2. UAV Multispectral Data

On the morning of 20 July 2021, a DJI Phantom 4 Pro V2.0 quadcopter drone (DJI, Shenzhen, China) (Figure 2a) equipped with a MicaSense RedEdge-Multispectral camera (MicaSense, Raleigh, NC, USA) (Figure 2b) was used to acquire data through five bands: blue (434–466 nm); green (544–576 nm); red (634–666 nm); red-edge (714–7746 nm); and near-infrared (824–856 nm). The remote control specified the range; the altitude was set to 150 m; the heading overlap rate and the side overlap rate were set to 79% and 66%, and then the flight path was automatically generated, its resolution reaching 10 cm.

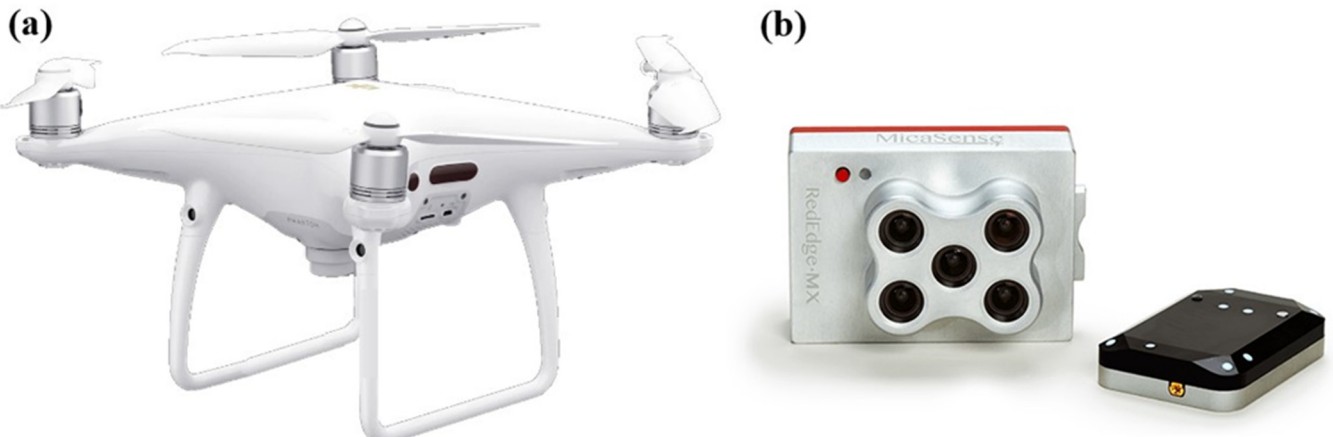

**Figure 2.** (**a**) DJI Phantom 4 Pro V2.0 quadcopter drone; (**b**) MicaSense RedEdge-Multispectral camera.

### 2.2.3. UAV LiDAR Data

On the afternoon of 22 July 2021, the point cloud data were collected using a Zhonghai Da six-rotor UAV Long-120 (Zhonghai DA, Guangzhou, China) (Figure 3a) equipped with an ARS-1000 L long-range LiDAR measurement system (Zhonghai DA, Guangzhou, China) (Figure 3b), and the main parameters are shown in Table 1. The UAV flew at an altitude of 200 m during the collection of the data, with flight speeds ranging from 6 to 10 m/s, a bypass overlap rate of 60%, and a heading overlap rate of 70%. The dispersion fraction of the LiDAR sensor beam was 0.5 rmad; the acquired data footprint diameter was between 0.1 m and 0.2 m, and the density of the point cloud was 70 points/m$^2$.

**Table 1.** Core parameters for LiDAR data acquisition.

| Project | Performance Parameters | |
|---|---|---|
| System Performance | Absolute precision | ±5 cm |
| | Weight | 4.5 kg |
| Laser Sweep Unit | Laser Safety Levels | Level 1 |
| | Maximum field of view | ±330° |
| | Angular resolution | 0.001° |
| | Data update rate | 200 Hz |
| | Maximum transmitting point frequency | 750 KHz |
| | Multi-echo function | 15 times |
| | Maximum range | 1350 m@60% |

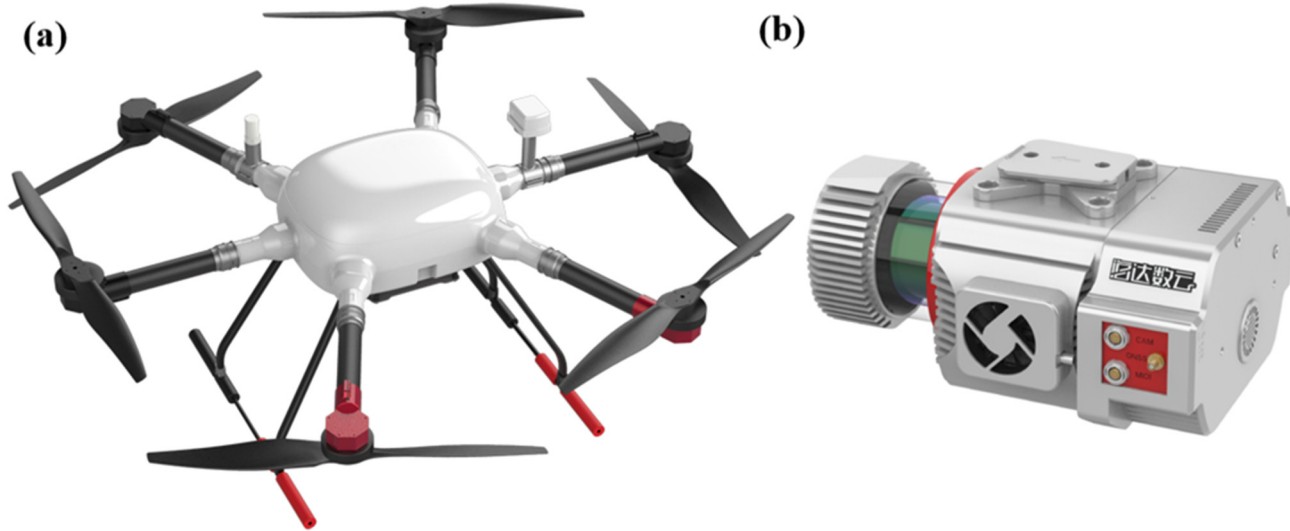

**Figure 3.** (**a**) Zhonghai Da six-rotor UAV Long-120; (**b**) ARS-1000 L long-range LiDAR measurement system.

### 2.3. Research Methods

### 2.3.1. Multispectral Vegetation Index Calculation

As the larch trees succumbed to insect pest stress, the internal biochemical components of the needles changed with increasing severity so that the canopy reflectance had obviously different responses [2]. Thus, the spectral information on each band could reflect the vegetation condition of different severities, so this study used the multispectral vegetation index as an indicator of insect pest identification. The area of interest was mapped by manual visual interpretation of the test area through ArcGIS10.7 software; 20 vegetation indices were calculated with the help of ENVI5.3 software using blue, green, red, red-edge, and near-infrared bands (see Table 2), and the average canopy reflectance of each sample tree was extracted as the value of the vegetation index for that tree.

**Table 2.** Vegetation indices and the basis of their calculation formulas.

| Number | Vegetation Index | Formulation | References |
|:---:|:---:|:---:|:---:|
| 1 | Anthocyanin Reflectance Index (ARI) | $1/B2 - 1/B4$ | Gitelson A A2009 [29] |
| 2 | Green Modified Simple Ratio (GMSR) | $(B5/B2 - 1)/(B5/B2 + 1)0.5$ | Knyazikhin Y1998 [30] |
| 3 | Green Normalized Difference Vegetation Index (GNDVI) | $(B5 - B2)/(B5 + B2)$ | Gitelson A A1996 [31] |
| 4 | Green Ratio Vegetation Index (GRVI) | $B5/B2$ | Gitelson A A2002 [32] |
| 5 | Modifies Nonlinear vegetation index (MNLI) | $1.5(B5^{0.5} - B3)/(B5^{0.5} + B3 + 0.5)$ | Peng G2003 [33] |
| 6 | Modified Simple Ratio (MSR) | $(B5/B3 - 1)/(B5/B3)^{0.5} + 1)$ | Philip N1982 [34] |
| 7 | Modified Simple Ratio–red edge (MSRreg) | $(B5/B4 - 1)/(B5/B4 + 1)^{0.5}$ | Chen J M1996 [35] |
| 8 | Normalized Difference Green Index (NDGI) | $(B2 - B3)/(B2 + B3)$ | Mirik M2012 [36] |
| 9 | Normalized Difference Salinity Index* (NDSI*) | $(B3 - B4)/(B3 + B4)$ | Richardson A D2002 [37] |
| 10 | Normalized Difference Salinity Index-Red Edge (NDSIreg) | $(B4 - B5)/(B4 + B5)$ | Rondeaux C1996 [38] |
| 11 | Normalized Difference Vegetation Index (NDVI) | $(B5 - B3)/(B5 + B3)$ | Rouse J W1974 [39] |
| 12 | Normalized Difference Vegetation Index* (NDVI*) | $(B4 - B3)/(B4 + B3)$ | Gitelson A 1994 [40] |

**Table 2.** *Cont.*

| Number | Vegetation Index | Formulation | References |
|--------|------------------|-------------|------------|
| 13 | Normalized Difference Vegetation Index-Red Edge (NDVIreg) | $(B5 - B4)/(B5 + B4)$ | Ortiz S M2013 [41] |
| 14 | Optimize Soil-adjusted Vegetation Index (OSAVI) | $(B5 - B3)/(B5 + B3 + 0.16)$ | Rondeaux C1996 [38] |
| 15 | Optimize Soil-adjusted Vegetation Index-Red Edge (OSAVIreg) | $(B5 - B4)/(B5 + B4 + 0.16)$ | Rondeaux C1996 [38] |
| 16 | Ratio Vegetation Index (RVI) | $B5/B3$ | Jordan C F1969 [42] |
| 17 | Ratio Vegetation Index* (RVI*) | $B4/B3$ | Major D J1990 [43] |
| 18 | Ratio Vegetation Index-Red Edge (RVIreg) | $B5/B4$ | Yang Ning 2020 [44] |
| 19 | Renormalized Difference Vegetation Index-Red Edge (RDVIreg) | $(B5 - B4)/(B5 + B4)^{0.5}$ | Broge N H2001 [45] |
| 20 | Wide Dynamic Range Vegetation Index (WDRVI) | $(0.1B5 - B3)/(0.1B5 + B3)$ | Gitelson A A2004 [46] |

### 2.3.2. LiDAR Feature Calculation

In this study, LiDAR360 6.0 (China Green Valley) software was used to process the acquired LiDAR point cloud data via resampling, denoising, ground point classification, and normalizing the data according to the ground points. An irregular triangular mesh interpolation was used to generate a digital elevation model (DEM) and a digital surface model (DSM) with a 1 m resolution, and then the difference between the DEM and the DSM was used to obtain a canopy height model (CHM). The CHM was used for the segmentation of the canopy, and all the segmented trees were used to map the spatial distribution of the severity of the larch caterpillar infestation in the experimental area. The accumulated interquartile height (elev_AIH) and height percentile (elev_Percentile) were obtained as another indicator of larch caterpillar infestation identification by calculating the forest parameters based on a 1 m × 1 m grid with LiDAR360 software. The accumulated interquartile height refers to the cumulative height of all normalized LiDAR point clouds within a given statistical unit sorted by height and the cumulative height of the points calculated. The cumulative height where X% of the points in each statistical unit are located is the cumulative height percentile of the statistical unit, which contains 15 intervals in LiDAR360, i.e., 1%, 5%, 10%, 20%, 25%, 30%, 40%, 50%, 60%, 70%, 75%, 80%, 90%, 95%, and 99%. The height percentile is the height of a statistical cell in which all normalized LiDAR point clouds within it are sorted by height, and then the height at which X% of the points within each statistical cell are located is calculated, which is the height percentile for that statistical cell, and the height percentile for the statistic likewise contains 15 intervals, i.e., 1%, 5%, 10%, 20%, 25%, 30%, 40%, 50%, 60%, 70%, 75%, 80%, 90%, 95%, and 99% (shown in Figure 4).

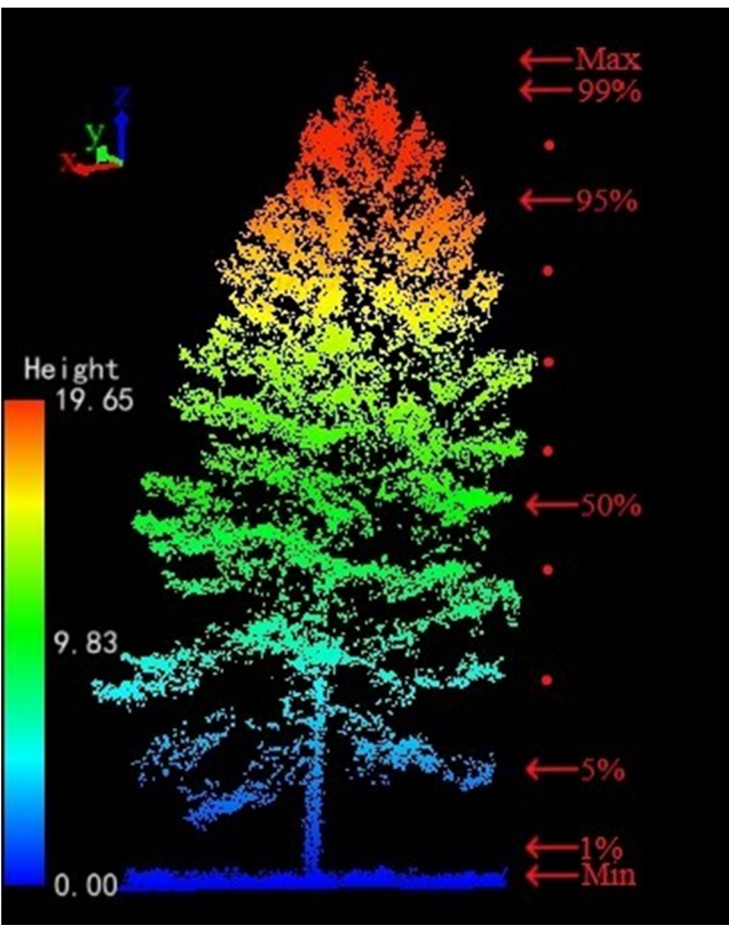

**Figure 4.** LiDAR accumulated interquartile height (elev_AIH) and height percentile (elev_ Percentile); 15 statistical variables.

### 2.3.3. Sensitive Feature Selection Method

Selection of multispectral vegetation indices and LiDAR-sensitive features is the most critical step before modeling, and using too many variables increases the computational time and complexity of the model and affects the model results, thus not always producing the best accuracy [47]. Therefore, in this study, the F test in ANOVA was utilized to reveal multispectral and LiDAR features sensitive to different severity levels caused by larch caterpillars. This method is a commonly used hypothesis test in ANOVA, where a threshold is used to compare whether there is a significant difference between between-group and within-group variations, with a larger F value indicating greater sensitivity [2].

### 2.3.4. Recognition Model

Based on sensitive multispectral and LiDAR features, MATLAB2022b software was used to implement Random Forest (RF) and Support Vector Machine (SVM) algorithms to construct identification models of larch caterpillar infestation with different severity.

The RF model is an integrated learning algorithm proposed by Leo Breiman and Adele Cutler, which is composed of multiple decision trees and a subset of training samples. In the training process of each of the multiple decision trees, a random subset of features and a subset of training samples are used to increase the model's generalization performance and resistance to noise [2]. Two important input parameters of this model are the number of classification trees (ntree) and the number of input variables for each split during tree construction (mtry), where ntree is set to 500, and mtry is set to 2 [41]. The SVM model is a learning method based on the criterion of minimizing structural risk, which has a better generalization ability than some traditional learning methods [48]. In MATLAB,

the RBF kernel function is used to solve the multi-classification problem, in which two important parameters, penalty factor (C) and Gamma coefficient (G), are found via the cross-validation method [49]. These two models have been widely used in remote sensing identification of forest pests and diseases and can effectively identify the distribution, scope, and development trend of forest pests and diseases.

2.3.5. Evaluation of Model Accuracy

To validate the accuracy of random forest and support vector machine models for monitoring larch caterpillar infestations at different severity levels, 285 out of 356 measured sample trees were used as the training set (80%), and 71 trees were used as the validation set (20%). In this paper, the overall accuracy (OA), kappa coefficient, overall recall ($R_{macro}$), overall F1 score ($F1_{macro}$), producer accuracy (PA), and user accuracy (UA) were selected to evaluate the accuracy of the predicted and actual results. OA was the ratio of the number of correctly categorized samples to the overall number of samples (see Equation (1)). The kappa coefficient is an index that combines user accuracy and producer accuracy, which can test the consistency between categorization results and field results, and if the kappa coefficient is larger, it indicates better consistency and vice versa (see Equation (2)). $R_{macro}$ and $F1_{macro}$ are indicators that extend the Recall and F1 scores and are extended into multicategory evaluation indices, which are then added and averaged. The same weight is given to each category, and $R_{macro}$ (see Equation (3)) and $F1_{macro}$ (see Equation (4)) are finally obtained; their values range from 0 to 1, with 1 indicating that the model is stable. Producer accuracy (PA) is the fraction of correctly categorized samples divided by the true situation, which can evaluate the model's ability to recognize each category, while user accuracy (UA) reflects the ratio of the number of correctly categorized samples of a category in the model's recognition results to the total sample tree of that category appearing in the recognition results, and these two metrics can be used to evaluate the accuracy and reliability of the recognition results.

$$OA = \frac{TP + TN}{(TP + TN + FP + FN)} \tag{1}$$

where OA is the overall accuracy; TP and TN are the numbers of sample trees correctly classified by the model, and FP and FN are the numbers of sample trees misclassified.

$$K = \frac{Po + Pe}{1 - Pe} \tag{2}$$

where K is the kappa coefficient; Po is the overall classification result, and Pe is the consistency error.

$$R_{macro} = \frac{1}{n} \sum_{i=1}^{n} R_i \tag{3}$$

where $R_{macro}$ is the total recall, and $R_i$ is the recall of class i.

$$F1_{macro} = \frac{1}{n} \sum_{i=1}^{n} F1_i \tag{4}$$

where $F1_{macro}$ is the overall F1 score, and $F1_i$ is the F1 score for category i.

The accuracy of all models was finally compared, and the identification model with the highest integrated accuracy was selected for mapping the spatial distribution of larch caterpillar infestation severities in the experimental area.

## 3. Results

### 3.1. Sensitivity Analysis of Multispectral Vegetation Indices and LiDAR Features

To select vegetation indices and LiDAR features that are sensitive to the severity of larch caterpillar infestation, the F variance values were calculated for 20 vegetation indices and 30 Li-

DAR features. For the multispectral vegetation indices, with F > 10.65 ($\alpha = 10^{-5}$), the sensitivity of ARI, MNLI, MSRreg, NDGI, NDSIreg, NDVIreg, OSAVI, OSAVIreg, RVIreg, and RDVIreg was strong, as seen in Figure 5a, with the largest F value of 88.79 for NDGI, and the worst sensitivity was GNDVI, reaching only 1.19. For LiDAR features, when F > 3.84 ($\alpha = 10^{-2}$), as seen in Figure 5b, the sensitivities of AIH_1st, pre_30th, _25th, pre_20th, pre_10th, and pre_1st, with the largest F value of 6.07 for pre_25th and the worst sensitivity being AIH_75th, reaching only 0.29. Therefore, the 10 multispectral vegetation indices and 6 LiDAR features mentioned above were finally selected as the input variables to construct the recognition model for different severity levels of larch caterpillar infestation.

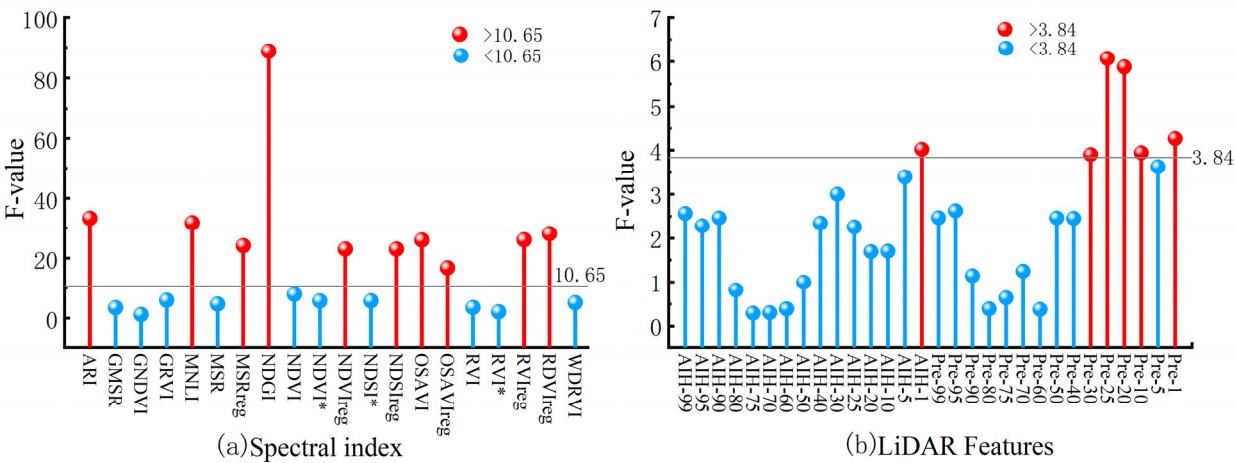

**Figure 5.** The variance of vegetation index and LiDAR features; (**a**) F-values for 20 vegetation indices; (**b**) F-values of the 30 LiDAR features, with red indicating highly sensitive features and blue indicating weakly sensitive features.

### 3.2. Larch Caterpillar Infestation Severity Recognition Model Results

To select the best-performing model for recognizing the severity of larch caterpillar infestation, six models were constructed by applying 10 sensitive vegetation indices and 6 sensitive LiDAR features, namely, random forest and support vector machine models based on 10 vegetation indices only ($RF_{VI}$ and $SVM_{VI}$); random forest and support vector machine models based on 6 LiDAR features only ($RF_{LIDAR}$ and $SVM_{LIDAR}$); random forest and support vector machine models based on 16 vegetation indices and LiDAR features ($RF_{VI+LIDAR}$ and $SVM_{VI+LIDAR}$); and after confounding the six models, the confusion matrices for the six models were created, as shown in Figures 6–8, where the green diagonal line denotes the number of correctly classified cases for the model, while the remaining red portion of the map is the number of incorrectly classified cases; the last row denotes the producer accuracy (PA), and the last column indicates the user accuracy (UA).

Figure 6 shows the confusion matrix results for $RF_{VI}$ and $SVM_{VI}$ based on only 10 multispectral features, with $RF_{VI}$ having an OA of 77.5%, while $SVM_{VI}$ has a better OA than RF at 90.1%. In producer accuracy, $RF_{VI}$ has the highest severe infestation accuracy (95.2%), and $SVM_{VI}$ has the highest mild infestation accuracy (100%), whereas in user accuracy, $RF_{VI}$ has the highest healthy tree accuracy (90.9%), and $SVM_{VI}$ has the highest severe infestation accuracy (100%). In the $RF_{VI}$ model, for the healthy canopy, six trees were misclassified as mild, and four trees were misclassified as severe; for mild infestation, two trees were misclassified as healthy, and three trees were misclassified as severe; for severe infestation, one tree was misclassified as mild. In the $SVM_{VI}$ model, for a healthy canopy, one tree was misclassified as mild; for mild infestation, misclassification did not occur; for severe infestation, four trees were misclassified as healthy, and two trees were misclassified as mild infestation. From Table 3, we know that the Kappa coefficient, $R_{macro}$, and $F1_{macro}$ values of $SVM_{VI}$ are higher than those of $RF_{VI}$, which are 0.8610, 0.9161, and 0.8947, respectively.

**Figure 6.** Recognition model of multispectral vegetation index confusion matrices results. (**a**) RF$_{VI}$; (**b**) SVM$_{VI}$.

**Figure 7.** Recognition model of LiDAR feature confusion matrices results. (**a**) RF$_{LIDAR}$; (**b**) SVM$_{LIDAR}$.

Figure 7 shows the results of the confusion matrices for RF$_{LIDAR}$ and SVM$_{LIDAR}$ based on only six LiDAR features, with SVM$_{LIDAR}$ reaching 62% OA while RF$_{LIDAR}$ reaches only 45.1% OA. The producer accuracy was again the highest for mild infestation (RF$_{LIDAR}$: 55%, SVM$_{LIDAR}$: 80%), and the user accuracy was also the highest for healthy trees (RF$_{LIDAR}$: 59.1%, SVM$_{LIDAR}$: 70%). In the RF$_{LIDAR}$ model, 5 healthy trees were misclassified as mild infested, and 12 trees were misclassified as severely infested; 3 mildly infested trees were misclassified as severely infested, and 6 trees were misclassified as severely infested; 6 severely infested trees were misclassified as healthy, and 7 trees were misclassified as mildly infested. In the SVM$_{LIDAR}$ model, for healthy trees, 6 trees were misclassified as mildly infested, and 10 trees were misclassified as severely infested; for mild infestation, 1 tree was misclassified as healthy, and 3 trees were misclassified as severely infested; for severe infestation, 5 trees were misclassified as healthy, and 2 trees were misclassified as mildly infested. As seen in Table 3, the Kappa coefficient, R$_{macro}$, and F1$_{macro}$ accuracies of SVM$_{LIDAR}$ are higher than those of RF$_{VI}$, which are 0.5509, 0.6284, and 0.6235, respectively.

**Figure 8.** Recognition model of fusing 16 vegetation indices and LiDAR features confusion matrix results. (**a**) $RF_{VI+LIDAR}$; (**b**) $SVM_{VI+LIDAR}$.

**Table 3.** Evaluation of the accuracy of classification models for different degrees of forest tree damage.

| Indicator \ Model | $RF_{VI}$ | $SVM_{VI}$ | $RF_{LIDAR}$ | $SVM_{LIDAR}$ | $RF_{VI+LIDAR}$ | $SVM_{VI+LIDAR}$ |
|---|---|---|---|---|---|---|
| OA | 0.7746 | 0.9014 | 0.4510 | 0.6200 | 0.7606 | **0.9577** |
| Kappa | 0.7150 | 0.8610 | 0.3748 | 0.5509 | 0.7012 | 0.9384 |
| $R_{macro}$ | 0.7772 | 0.9161 | 0.4590 | 0.6284 | 0.7658 | 0.9595 |
| $F1_{macro}$ | 0.7723 | 0.8947 | 0.4507 | 0.6235 | 0.7601 | 0.9594 |

Figure 8 represents the confusion matrix results of $RF_{VI+LIDAR}$ and $SVM_{VI+LIDAR}$ fusing 16 vegetation indices and LiDAR features, where $SVM_{VI+LIDAR}$ showed the highest overall accuracy (OA = 95.8%) and the other three evaluation metrics were higher than 0.9, as seen in Table 3; $RF_{VI+LIDAR}$ had an OA of 76.1%, and the Kappa coefficient, $R_{macro}$, and $F1_{macro}$ reached 0.7012, 0.7658, and 0.7601, respectively. In producer accuracy, $RF_{VI+LIDAR}$ has the highest accuracy in mild infestations (85%), and $SVM_{VI+LIDAR}$ has the highest accuracy in severe infestations (100%). For user accuracy, $RF_{VI+LIDAR}$ has the highest accuracy in severe infestations (90.9%), and $SVM_{VI+LIDAR}$ has the highest accuracy in mild infestations (100%). In the $RF_{VI+LIDAR}$ model, four healthy trees are misclassified as mildly infested; six trees are misclassified as severely infested; two mild trees are misclassified as healthy; one tree is misclassified as severely infested, and four severe trees are misclassified as mildly infested. In the $SVM_{VI+LIDAR}$ model, two healthy trees were misclassified as severely infested; one mildly infested tree was misclassified as healthy, and no misclassification occurred for severely infested trees.

In summary, the combined accuracy of $SVM_{VI+LIDAR}$ was the highest, reaching OA: 0.9577, Kappa: 0.9384, $R_{macro}$: 0.9595, and $F1_{macro}$: 0.9594. Although the accuracy of the SVM model based on the vegetation index alone also reached more than 90%, the overall accuracy of the $SVM_{VI+LIDAR}$ model increased by 5.63% with the addition of the LiDAR feature, indicating that combining the vegetation index and LiDAR feature to monitor the severity of larch caterpillar infestation would yield results closer to reality; $RF_{LIDAR}$ had the lowest combined accuracy of OA: 0.4510, Kappa: 0.3748, $R_{macro}$: 0.4590, and $F1_{macro}$: 0.4507, suggesting that the use of LiDAR AIH and height percentile alone for monitoring the severity of larch caterpillar infestations could not achieve a reliable accuracy. For the SVM model, the accuracies of all scoring metrics, whether based on the vegetation index,

LiDAR features, or a combination of the two, were higher than those of the RF model, which showed that the SVM model had a strong ability to generalize because the combined accuracy was high due to the 10-fold cross-validation method applied to the training and validation sets when using this model. For the RF model with three input variables, the $RF_{VI}$ had the highest integrated accuracy of OA: 0.7746, kappa: 0.7150, $R_{macro}$: 0.7772, and $F1_{macro}$: 0.7723, which indicated that the random forest model was more suitable for monitoring larch caterpillar infestations based on the multispectral vegetation index. In summary, $SVM_{VI+LIDAR}$, which possesses the best-integrated accuracy, was finally selected for mapping the spatial distribution of pine caterpillar infestation severity in the study area segmented by the LiDAR CHM.

### 3.3. Distribution of Larch Caterpillar Infestation Severity Based on the Single-Plant Scale

To map the spatial distribution of larch caterpillar infestation severity in the test area, 7879 trees were obtained from single-tree segmentation by the LiDAR canopy height (CHM) model in this study. The sensitive multispectral vegetation index and LiDAR features were calculated for the acquired single trees and then input into the identification model with the highest accuracy ($SVM_{VI+LIDAR}$). As previous research indicated that insect infestation is related to topographic features [50], elevation information was extracted from the DEM data of the experimental area at the monoculture scale. As seen in Figure 9a, the identification model classified 5560 healthy, 2145 mildly infested, and 174 severely infested trees, which shows that this experimental area is dominated by a healthy canopy, concentrated in the northeast, accounting for 70.6% of the total trees; mildly infested stands accounted for 27.2% of the total trees, mainly located in the southwest, and severely infested stands accounted for only 2.2% of the total trees, most of which were distributed in the southwest, while a few were found throughout the northeast. As shown in Figure 9b, the terrain in the north-central part of the experimental area was high, and the terrain became flatter as it continued southward, with the lowest elevation of 715.90 m and the highest elevation of 799.34 m. As seen in Figure 9c, the mildly and severely infested canopies were mainly concentrated in the areas with elevations of 748–758 m and 737–747 m. In conclusion, the severity of larch caterpillar infestation in this test area increases with decreasing elevation, and timely control should be carried out at low elevations to avoid seriously affecting healthy crowns in high elevations.

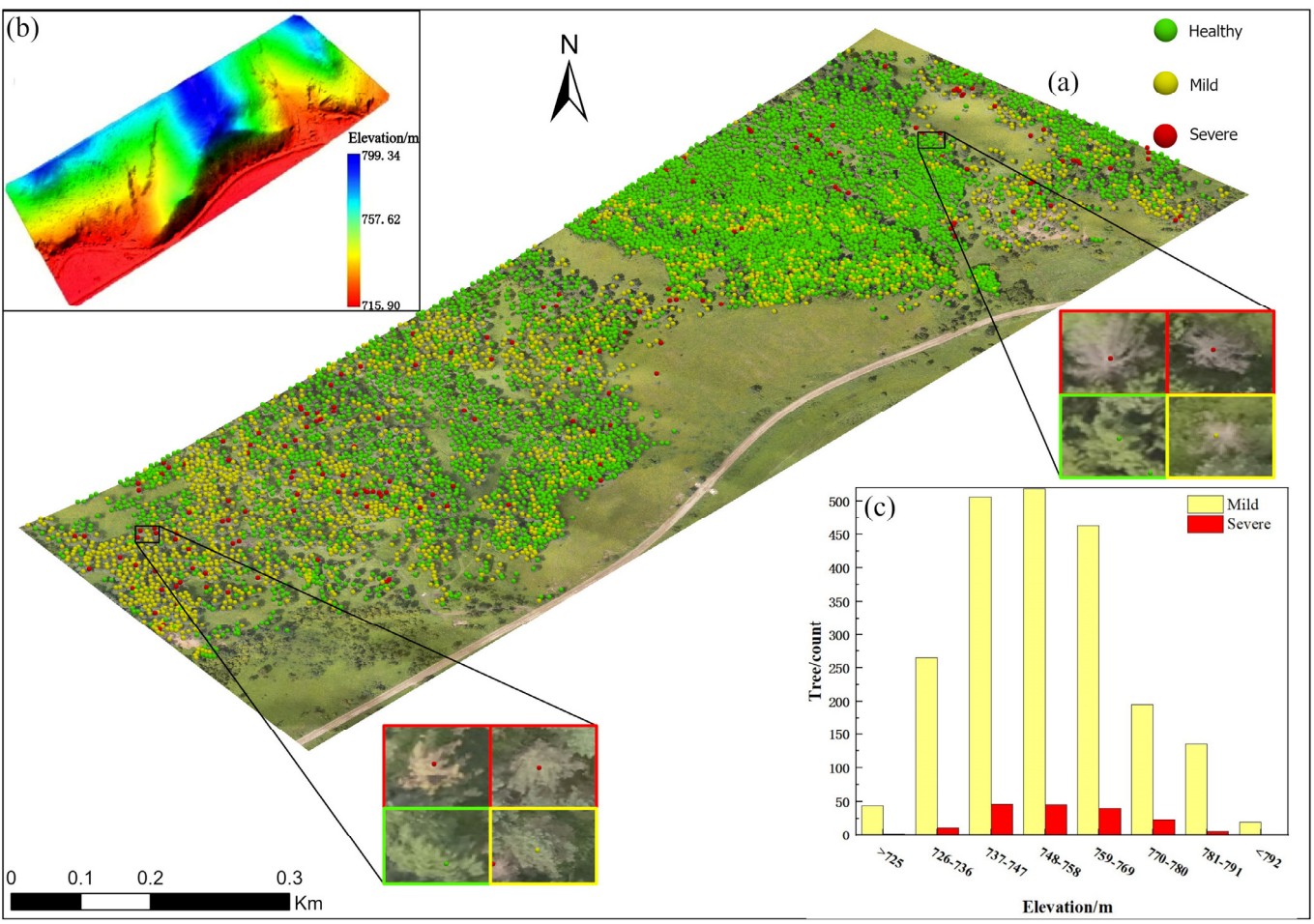

**Figure 9.** (**a**) Spatial distribution and local magnification map of larch caterpillar infestation severity at single plant scale in the study area based on $SVM_{VI+LIDAR}$ model; (**b**) DEM of the study area; (**c**) Histogram of pest severity at different elevations.

## 4. Discussion

### 4.1. Multispectral Vegetation Indices and Sensitivity of LiDAR Features

Larch trees under pine caterpillar infestation stress undergo significant changes in their exterior color and internal biochemical components, thus allowing for the identification of different levels of severity using multispectral vegetation indices and LiDAR features. Vegetation indices are calculated by different combinations of visible and near-infrared wavelengths [51] and have different canopy reflectances in each wavelength when the forest is attacked, so it is especially critical to extract vegetation indices that are sensitive to different severities [52]. In this paper, F values were calculated for 20 indices, and 10 sensitive vegetation indices were obtained by thresholding, with the most sensitive NDGI index being the normalized differential greenness index integrated through the green and red bands, which was initially defined for the measurement of structural properties of vegetation, correlated with greenness measurements [53], and used to test for different vigorous vegetation forms [54]. When larch trees are stressed by larch caterpillars, the color of the canopy changes from green to grey, and thus, the green reflectance of the spectrum is weakened, whereas in the red band, with the increase in the leaf loss rate (LLR), the chlorophyll and water content of the needles decreases, leading to an increase in reflectance. Therefore, this index has a strong correlation with insect pest stress, and it can be used as an important multispectral vegetation index for monitoring larch caterpillars. The six (MSRreg, NDSIreg, NDVIreg, OSAVIreg, RVIreg, RDVIreg) subindices in the sensitivity index were derived by replacing the red band calculated by the original MSR, NDSI, NDVI, OSAVI, RVI, and RDVI with the red edge band [55], where the reflectance of the red band increases with decrease in leaf area; the near-infrared (NIR) band

has a strong reflective characteristic on the cellular tissues, and as the severity of damage to the cellular tissues of the larch needles increases the reflectivity decreases. The OSAVI and MNLI are vegetation indices integrated by combining the green and red-edge bands. Among them, OSAVI is an optimization of the SAVI index, establishing 0.16 as the optimal soil adjustment factor, which minimizes the variation with soil background and is more sensitive to vegetation health [55]. Similarly, the MNLI [33], a vegetation index obtained by modifying the original nonlinear vegetation index (NLI) with soil factors, is strongly correlated with the leaf area index (LAI). The decrease in LAI due to needle consumption by larch caterpillars is more sensitive to vegetation health [56]. ARI, an anthocyanin reflectance index calculated as the difference between the green and red-edge band inversions, is particularly sensitive to anthocyanin in needles and leaves. Increasing severity changes the color of needles as chlorophyll converts to anthocyanin, so the accumulation of anthocyanin leads to enhanced reflectance [57]. Figure 10a shows the normalized values of the sensitive multispectral vegetation indices at different severities, where the normalized values of the indices ARI, MSRreg, NDVIreg, OSAVIreg, and RVIreg decrease significantly with increasing severity, thus making it easier to distinguish between different severities. The MNLI and OSAVI indices had increasing and then decreasing normalized values on mildly damaged canopies, while the RDVIreg indices all had smaller normalized values than the others. The least sensitive index is the green normalized difference vegetation index (GNDVI), calculated from the NIR and green bands. From many studies, it has been found that the GNDVI is mostly used for estimating crop biomass and chlorophyll based on UAV multispectral imagery as well as identifying low-nitrogen stress and monitoring aquatic organisms [58–60] and is, therefore, less sensitive in terms of forest pest and disease stress monitoring.

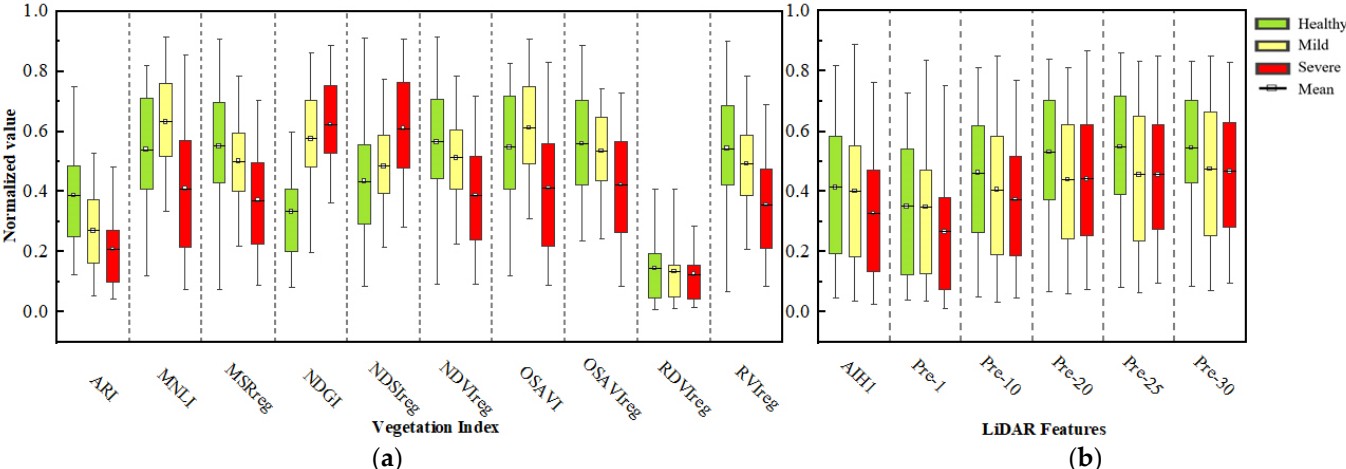

**Figure 10.** Normalized values of sensitivity features of different severity. (**a**) Multispectral vegetation index; (**b**) LiDAR feature.

Lidar features are less sensitive to health status due to the difficulty of measuring changes in biochemical components within the stand but can be good for monitoring stand growth structure in the vertical direction. In studies of tree species identification and estimation of parameters such as tree height and volume, 80%–90% of the height variables can represent the dominant height of the trees in the sample plot and become the main determinants of species identification as well as estimation of parameters [61]. On the contrary, the LiDAR-sensitive features extracted in this study mainly described the appearance of the lower and middle layers of the trees (pre_30th, pre_25th, precentile_20th, pre_10th, pre_1st, and AIH_1st), which was explained by the fact that the larvae of the larch caterpillar overwintered under the dead branches and leaves, and then infested the larch from the bottom up in the second year, threatening first the branches in the lower and middle layers of the forest [62], so the sensitive characteristics coincide with the most obvious parts of the infested forest. Figure 10b shows the normalized values of the sensitive

LiDAR features at different severities, and it was found that all six sensitive features showed a trend of decreasing normalized values with increasing severity, with insignificant changes compared to the multispectral vegetation index. From the normalized mean values, the AIH1 and Pre-1 features could not effectively differentiate between healthy and mildly damaged canopies, while Pre-20, Pre-25, and Pre-30 were poor at identifying mildly and severely damaged canopies. The severity of infection with the invasive Phytophthora ramorum in larch has also been categorized by the metrics of canopy cover, skewness, and height variables from airborne LiDAR (ALS) point clouds, and it has been demonstrated that ALS can be used to characterize individual canopies of moderately and severely infected trees [63]. In the next study, more vegetation indices combined with near-infrared and red-edge bands will be calculated in conjunction with forest structural parameters such as Li-DAR leaf area index, clearance rate, biomass, and intensity variables, and attempts will be made to monitor forest pests and diseases in multidimensional space using a variety of susceptibility extraction methods to visualize the results of the study in three dimensions.

*4.2. Accuracy of Larch Caterpillar Pest Severity Monitoring Models*

In this study, we aimed to construct six larch caterpillar pest severity identification models based on random forest and support vector machine algorithms using multispectral vegetation index and LiDAR features combined with ground survey data. As is widely known, traditional machine learning and deep learning models are commonly used research methods in forest pest monitoring. Studies have shown that deep learning is better at classifying, detecting, and recognizing pests compared to traditional machine learning. However, it requires a large amount of training data and the support of high-performance computer hardware [16,64]. Therefore, in this study, we chose to use the most frequently used models of Random Forests and Support Vector Machines for our research [65,66]. Scholars compared deep learning (3D convolutional neural networks, backpropagation neural networks, artificial neural networks) and traditional machine learning (support vector machines, random forests, decision trees, and plain Bayes) in tree classification and disease severity classification using UAV hyperspectral, multispectral, and LiDAR data. The recognition accuracy of the support vector machine model (SVM) of the traditional machine learning algorithms is more similar to that of deep learning, with the difference in recognition accuracy ranging from 0.27% to 2.6% [16,67]. Recently, other scholars have applied SVM and RF models to classify the severity of cotton diseases and oak leaf-eating insects obtained from unmanned aerial multispectral imagery, all obtaining more than 90% overall accuracy [68,69]. The support vector machine model constructed in this study based on multispectral vegetation indices and LiDAR features showed optimal overall accuracy (OA = 95.8%). This result is consistent with previous studies, indicating that the fusion of multisource remote sensing data can effectively improve the accuracy of the recognition model [16,18,66,70].

In this study, PA and UA of the $SVM_{VI+LIDAR}$ model for the identification of healthy, mild, and severe damaged canopies reached more than 90% (Figure 11), which shows that the larch caterpillar infestation identification model constructed on the basis of the 16 sensitive features extracted from the variance provides a reliable reference for identifying larch caterpillar infestations in forest trees of varying severities. Among the six recognition models, PA and UA based on LIDAR features alone were overall lower as vegetation index and a combination of both, with $SVM_{LIDAR}$ having higher accuracy than $RF_{LIDAR}$. Although the LIDAR height variable used in this paper did not accurately identify pest severity, other studies have shown that combining the LODAR intensity variable with hyperspectral data can detect the stress aspect of pine tip beetle with high accuracy ($R^2 = 0.83$, RMSE = 15.84%) [18]. $SVM_{VI}$ has a PA rate of 100% in identifying mildly damaged crowns. Additionally, it is capable of distinguishing subtle differences between healthy and mildly damaged trees through the use of multispectral data, which can monitor changes in both the external condition of trees and their internal biochemical components [71]. The $SVM_{VI+LIDAR}$ model achieved a 100% PA in identifying severely damaged canopies. This

was due to the fact that most needles of the severely damaged canopies in the experimental area were shed, and all branches were exposed, resulting in large differences in canopy reflectance and vertical structure information at different heights compared to healthy and mild damaged stands [72]. Future attempts will include the use of multisource remote sensing data combined with UAV multispectral and hyperspectral data as well as backpack LiDAR data to mine a more accurate identification model of larch caterpillar infestations of different severities and to achieve three-dimensional visualization of the research results.

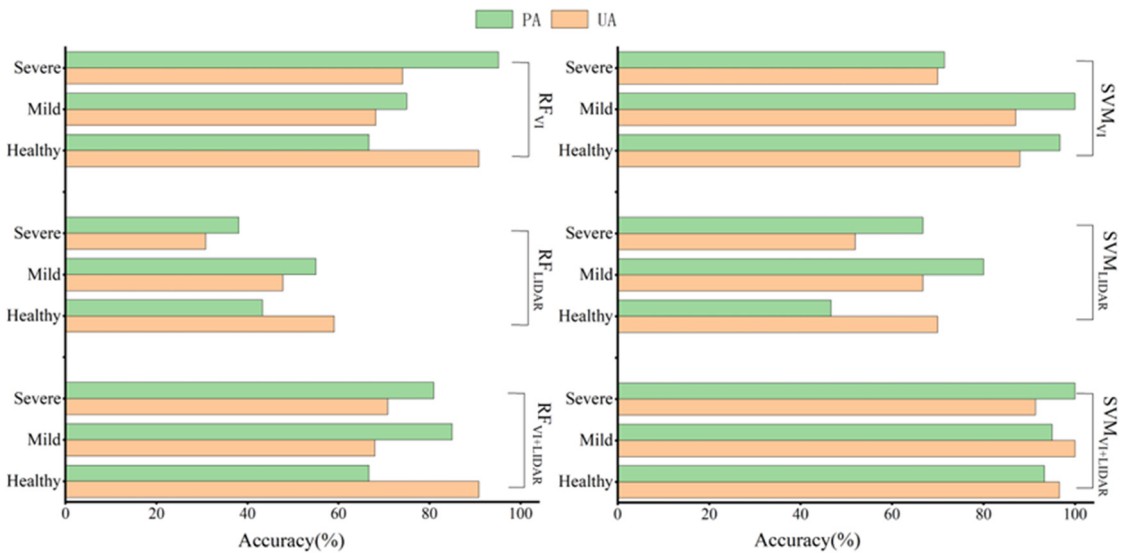

**Figure 11.** PA and UA of RF and SVM models with different feature sets.

## 5. Conclusions

In this study, we used UAV multispectral and LiDAR data combined with field measurement data to select vegetation indices, and LiDAR features sensitive to different severities of larch caterpillar infestation using the F test, constructed an effective pest severity identification model with the help of random forest (RF) and support vector machines (SVM) and used the model with the best integrated accuracy to map the severity of larch caterpillar infestation at the single-plant scale and interpret the spatial distribution characteristics according to topographic features. The conclusions are as follows:

(1) Ten multispectral vegetation indices and six LiDAR features were selected by the ANOVA test, of which the strongest sensitivities were the NDGI and Pre_25%. The vegetation indices calculated from the NIR and red-edge bands accounted for the largest number (6/10); the strong sensitivity of the 25% height variable was correlated with the date of collection of the experimental data and the growth cycle of larch caterpillars;

(2) Among the six monitoring models with different feature sets derived from the study, the $SVM_{VI+LIDAR}$ model has the highest integrated accuracy, with OA, KAPPA, $R_{macro}$, and $F1_{macro}$ above 0.95, and the overall accuracy of the model is improved by 5.63% and 33.77% compared with $SVM_{VI}$ and $SVM_{LIDAR}$, respectively. It can be seen that Multisource remote sensing data synergy is an important way to improve the accuracy of pest recognition;

(3) A high-precision monitoring model ($SVM_{VI+LIDAR}$) was used to map the severity distribution of larch caterpillar infestation in the study area based on a single-plant scale, and the trend of infestation was analyzed according to the topographic characteristics. It was found that the severity of larch caterpillar infestation tended to increase with decreasing elevation and that control should be carried out first on mildly and severely damaged canopies at low elevations to protect healthy canopies at high elevations.

At present, this paper provides a high-precision monitoring model for the identification of different severities of larch caterpillar infestations, which provides a reliable technical means for the control of forest pests in Great Khingan.

**Author Contributions:** S.H.-Y., X.H. analyzed the data and wrote this paper; X.H. conceived and designed these experiments; D.Z., J.Z. provided test site and measurement data; G.B., S.T., Y.B., D.G., N.T., D.A., D.E., M.A. and J.G. revised this manuscript. All authors have read and agreed to the published version of the manuscript.

**Funding:** This study was supported by the National Natural Science Foundation of China (42361057), the Inner Mongolia Autonomous Region Science and Technology Plan Project (2021GG0183), the Natural Science Foundation of Inner Mongolia Autonomous Region (2022MS04005), the Young Scientific and Technological Talents in High Schools (NJYT22030), the Ministry of Education Industry–University Cooperative Education Project (202102204002).

**Data Availability Statement:** Data are contained within the article.

**Conflicts of Interest:** The authors declare no conflict of interest.

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
