# Peer review of "Identification of Larch Caterpillar Infestation Severity Based on Unmanned Aerial Vehicle Multispectral and LiDAR Features"

_forests, doi:10.3390/f15010191_

Round 1

Reviewer 1 Report

Comments and Suggestions for Authors

L:20-21: Recently, forest pests, mainly larch caterpillars, have inflicted significant damage on the larch trees in the Great Khingan forest area. This statement is specific, while the preceding two sentences discuss a global situation. Please ensure consistency.

L40: The research title is somewhat general, yet the introduction begins with a very specific case study. Consider revising for better alignment between the title and the introduction.

L43- 57: Please include the necessary citation.

L77-78: The statement regarding hyperspectral data appears to be incorrect.

L79-81: The authors compare multispectral and hyperspectral data, asserting that multispectral is more powerful. However, for early detection of insect infestation or stress, hyperspectral bands are generally more effective due to their sensitivity to stress indicators not present in multispectral data. Consider revising this section to present a more accurate narrative.

Figure 1: I recommend modifying the map to overlay the study area based on surrounding locations for clarity in understanding its location.

L146-147: Include accuracy information for the utilized GPS.

L168-169: If the flight height is set to 150m, specify the software used to design the flight routes. Different software may behave variably at this consistent height, and it's crucial to note potential limitations.

L225: While the authors used the ANOVA test, it is unclear whether the data's distribution was tested for normality prior to this. Consider clarifying this point. Additionally, why not use Random Forest or PLSR to select important variables instead of relying solely on classical tests?

Figure 5: Enhance the quality of this figure.

Figure 9: The zoomed area does not provide meaningful information and makes it challenging to interpret what is beneath the circles. Consider improving clarity in this section.

L454-473: This paragraph needs citation and should justify results based on previous studies or make comparisons with existing literature.

L477-549: This entire section requires modification to incorporate and compare the main findings of this study with those of previous studies. The current form resembles a results section and lacks the necessary contextualization.

Reviewer 2 Report

Comments and Suggestions for Authors

The manuscript describes the usability of UAV multispectral, LiDAR data and ground survey to identify healthy, 

mild and severe larch caterpillar infestations. 

Generally the structure is appropriate, no new methodological approach is proposed, 

the focus falls on findings, results and conclusions regarding the aplicability of UAV Multispectral and LiDAR 

for identification of larch caterpillar infestation severity.

The adopted subject is appropriate and raises interest for the readers of the journal, however there are some points and missing 

information that needs to be covered:

Point 1. When the LiDAR point density is 70 points / square meter, why the derived DSM and DTM are computed at 1 m resolution? 

This decreses the quality of the models. Although they were employed for nDSM computation, the object height accuracy at 1m resolution is limited. An option is to compute the models at a higher resolution , ex. 10 cm.

Point 2. Why for the experiments these two models were speciffically chosen: RF and SVM when new deep learning models are available?

Point 3. To Prove the conclusions, the adopted methodology should be applied on other datasets with other characteristics, as well, then the conclusions

are not limited to this specific test study area of Great Kingham Forest.

Point 4: Figures ,8, 9: "confusion matrices"

Point 5: How are the multispectral images and LiDAR data combined and what are the accuracy metrics for the georeferencing?

Point 6: How are the SVM and RF models implemented? Please give more details about the processing steps

Comments on the Quality of English Language

English quality presentation is fine.

Reviewer 3 Report

Comments and Suggestions for Authors

No more comment.

Author Response

Thank you for your review.

Round 2

Reviewer 1 Report

Comments and Suggestions for Authors

I'm satisfied with the author's reply to my comments and the modifications they made in the manuscript